# BiP Proteins from Symbiodiniaceae: A “Shocking” Story

**DOI:** 10.3390/microorganisms12112126

**Published:** 2024-10-23

**Authors:** Estefanía Morales-Ruiz, Tania Islas-Flores, Marco A. Villanueva

**Affiliations:** Instituto de Ciencias del Mar y Limnología, Unidad Académica de Sistemas Arrecifales, Universidad Nacional Autónoma de México-UNAM, Prolongación Avenida Niños Héroes S/N, Puerto Morelos 77580, Quintana Roo, Mexico; esmoru@gmail.com (E.M.-R.); tislasf@gmail.com (T.I.-F.)

**Keywords:** BiP, chaperone, dinoflagellates, light-modulation, phosphorylation, Symbiodiniaceae

## Abstract

More than four decades ago, the discovery of a companion protein of immunoglobulins in myeloma cells and soon after, of their ability to associate with heavy chains, made the term immunoglobulin binding protein (BiP) emerge, prompting a tremendous amount of effort to understand their versatile cellular functions. BiPs belong to the heat shock protein (Hsp) 70 family and are crucial for protein folding and cellular stress responses. While extensively studied in model organisms such as *Chlamydomonas*, their roles in dinoflagellates, especially in photosynthetic Symbiodiniaceae, remain largely underexplored. Given the importance of Symbiodiniaceae-cnidarian symbiosis, critical for the sustaining of coral reef ecosystems, understanding the contribution of Hsps to stress resilience is essential; however, most studies have focused on Hsps in general but none on BiPs. Moreover, despite the critical role of light in the physiology of these organisms, research on light effects on BiPs from Symbiodiniaceae has also been limited. This review synthesizes the current knowledge from the literature and sequence data, which reveals a high degree of BiP conservation at the gene, protein, and structural levels in Symbiodiniaceae and other dinoflagellates. Additionally, we show the existence of a potential link between circadian clocks and BiP regulation, which would add another level of regulatory complexity. The evolutionary relationship among dinoflagellates overall suggests conserved functions and regulatory mechanisms, albeit expecting confirmation by experimental validation. Finally, our analysis also highlights the significant knowledge gap and underscores the need for further studies focusing on gene and protein regulation, promoter architecture, and structural conservation of Symbiodiniaceae and dinoglagellate BiPs in general. These will deepen our understanding of the role of BiPs in the Symbiodiniaceae-cnidarian interactions and dinoflagellate physiology.

## 1. Introduction

### 1.1. Discovery of BiP Proteins

In 1975, Morrison and Scharf [1] reported a companion 77 kDa protein that co-precipitated with immunoglobulin heavy chains synthesized by myeloma cell lines. This companion protein was chemically unrelated to the immunoglobulin heavy chains. Subsequently, the binding ability of this 77–78 kDa protein to immunoglobulin was discovered in pre-B lymphocytes and hybridomas, and the term immunoglobulin heavy-chain binding protein, or BiP, was coined [2]. BiP was shown to interact noncovalently with the heavy chains, and it was suggested that it might regulate the immunoglobulin chain synthesis [2]. Later, the same authors proposed that BiPs may constitute an antidote for the toxic effect of immunoglobulin chains [3]. In the same decade, Lee et al. [4] characterized a 78 kDa glucose-regulated protein (GRP78) and showed that its N-terminus was similar to that of the 72–73 kDa heat shock proteins from HeLa cells. That same GRP78 protein was cloned by Munro and Pelham in 1986 [5] and concluded that it was identical to BiP. Bole et al. [6] then showed that BiPs were involved in the posttranslational processing of nascent immunoglobulin heavy chains in the endoplasmic reticulum. After cloning and sequencing the mouse BiP cDNA, it was demonstrated that the 75 kDa BiP protein sequence matched that of the 70 kDa heat shock protein (Hsp70) family [7]. It was soon afterwards discovered that BiP and BiP-like proteins were fundamental for a variety of cellular processes and that they were present in different organisms since they are members of the 70 kDa heat shock response protein (Hsp70) family.

### 1.2. Functional Insights of BiP Proteins

It is nowadays clear that Hsp70 proteins play a critical role in maintaining the stability and proper functioning of cells in response to stress. They do this by assisting in the folding and assembly of newly synthesized proteins in the endoplasmic reticulum (ER), as well as the repair and refolding of damaged proteins. The proper folding of proteins is essential for their biological function since proteins are involved in almost every biological process. Chaperones assist protein folding so that the nascent polypeptides can assemble into their native three-dimensional structures. They also prevent misfolding by inhibiting protein aggregation or mediating targeted disassembly [8]. BiP proteins have also been suggested to play a critical role in the Unfolded Protein Response (UPR) at the ER, which is a signaling system that detects and responds to misfolded proteins at the ER [9]. This critical role ensures the survival of the organism in the face of environmental stressors, such as changes in temperature or exposure to toxins [10]. Hsp70 proteins also play a role in the response to other types of stress, such as changes in osmotic pressure and changes in nutrient availability. These proteins are found in all organisms and are highly conserved across different species [11]. The challenging conditions in which BiP proteins work suggest a fine modulation mechanism for their functions through post-translational modifications (PTMs) such as phosphorylation events [12,13,14,15,16,17]. In fact, Hsp70 proteins can be phosphorylated on several serine and threonine residues by various kinases, which suggests that these proteins are the target of different signaling pathways [12,15,16,17]. The phosphorylation of Hsp70 can have both positive and negative effects on its function and is thought to modulate its binding affinity with clients and other co-chaperones [16,17]. Despite the importance of these proteins in protein synthesis and assembly, very little is known about their biochemical properties and functions in dinoflagellates or other phylogenetically related organisms, including Symbiodiniaceae.

According to the World Register of Marine Species, dinoflagellates are protists from the phylum Myzozoa and further classified into a subphylum Dinozoa and the infraphylum Dinoflagellata [18]. They are related to other members of the alveolates, such as ciliates and apicomplexans. This is also reflected in their morphological diversity and varied lifestyles that include photoautotrophy, phagotrophy, mixotrophy, and parasitism [19]. Among the Dinoflagellata, a peculiar group of photosynthetic dinoflagellates of the Symbiodiniaceae family wrongly termed as “algae” stands out. That is because Symbiodiniaceae, also known as “zooxanthellae”, are a diverse family capable of living free in the ocean water column but also of establishing a symbiotic relationship with a wide range of marine organisms, including cnidarians such as reef-building corals [20]. Thus, they are particularly interesting due to their importance to sustain healthy coral reefs and, thereby, one of the most widely studied photosynthetic dinoflagellate families. Among the many important functions these symbionts perform, a most critical one is the ability to photosynthesize and provide energy to their host [20]. This symbiotic relationship can terminate due to environmental stressors such as changes in temperature and salinity. This causes the zooxanthellae to abandon their host, thus leading to coral mortality in a phenomenon known as bleaching (due to the lack of the photosynthetic brown pigments of the zooxanthellae when absent in the white coral skeleton) [20].

Since elevated temperatures can trigger the activity of the Hsps, it is possible that these proteins play a central role in the response of Symbiodiniaceae to stress. In fact, the presence of Hsp70 and Hsp90 chaperones is considered an important evolutionary adaptation arising from early endosymbioses between eubacteria and archaebacteria to give rise to eukaryotes [21], and further similar adaptations in other symbiotic relationships have likely occurred during evolution. Although studies on other chaperones of the Hsp70 family have been carried out previously [22,23,24,25], only recently biochemical properties and possible functions of BiP-like proteins from Symbiodiniaceae have been reported [12,13,14,15]. In this review, we discuss the current state of knowledge on the structure and function of BiP proteins from Symbiodiniaceae, as well as their evolutionary aspects and their potential implications for the survival and health of coral reefs. We also discuss what is currently known about BiP proteins in other dinoflagellates.

## 2. Stress Responses, Light-Regulated Phosphorylation, and Possible Roles of BiP Proteins in Symbiodiniaceae

In the context of Symbiodiniaceae, Hsps are particularly significant due to their abovementioned symbiotic relationships with coral hosts. Thus, environmental stress conditions such as elevated temperatures must exert a profound effect on Symbiodiniaceae Hsps, as it occurs in most eukaryotic organisms [26]. Of particular interest are the BiP proteins from the Hsp family, largely undescribed and uncharacterized in these organisms despite their significance in crucial biological processes [13,16,17,26,27].

The exact molecular interactions of BiP chaperones of the Hsp70 family with client proteins in Symbiodiniaceae are still far from understood; furthermore, only one group has reported and studied a BiP chaperone from Symbiodiniaceae to certain detail [12,13,14,15]. Most studies have been carried out documenting up-regulation of Hsp70 family chaperone genes leading to a protective effect against stress on other photosynthetic organisms [28]. Consequently, since heat stress primarily affects photosynthetic function in Symbiodiniaceae [29,30,31,32,33], it was assumed that Hsp chaperones in Symbiodiniaceae were likely to have important implications for the survival and health of coral reefs. Logically, then, they were targeted as relevant stress-responsive proteins for study in these organisms [22,23]. However, down-regulation of Hsp70 expression levels [22] or no significant differences [23,25] were observed between 32–34 °C heat stressed and normal temperature-growing Symbiodiniaceae. On the other hand, Levin et al. [24] reported that transcriptome analysis of a known thermotolerant cultured type C1 Symbiodiniaceae showed up-regulation of various molecular chaperone genes (including Hsp70) by ≥ 4-fold when grown at 32 °C compared to the expression of the same genes at normal growth temperature of 27 °C; however, none corresponded to BiP-type chaperones. In fact, none of the reported previous studies were directed towards BiP chaperones of the Hsp70 family.

The first and so far, only report of a *bona fide* BiP-like protein from Symbiodiniaceae was from Castillo-Medina et al. [12]. The protein was initially identified as a 75 kDa polypeptide that was significantly phosphorylated on Thr under darkness but rapidly dephosphorylated after exposure to light. Peptide sequencing identified the protein as a BiP-like protein of the Hsp70 family initially named SmicHSP75. The light modulated Thr phosphorylation/dephosphorylation response was observed in three different Symbiodiniaceae species, suggesting a common signaling pathway among them and not a species-specific light response mechanism. Indeed, further analysis showed the presence of at least five sequences homologous to Hsp70 family proteins in an annotated *S. microadriaticum* transcriptome [34], of which two were typical BiP-like proteins [35] (Figure 1). It was later shown that the reported BiP-like protein from *S. microadriaticum* was misannotated, and the correct sequence is now available in the GenBank with accession number OP429595.1 [14]. A search in the available databases revealed numerous Symbiodiniaceae BiP-like homologous sequences (Table 1), and the protein was named SBiP1 for Symbiodiniaceae BiP number 1 [13]. Furthermore, the overall protein sequence also displayed the HDEL C-terminal sequence indicating ER residence typical of BiP proteins (Figure 1 and Figure 2B, black boxed and C-terminal motif, respectively), indicating that it was an ER-resident type chaperone [14]. 

As mentioned above, homologous BiP sequences were found present across the various genomic and proteomic databases [14] (Table 1); however, no other BiP-like protein has been reported and/or characterized for Symbiodiniaceae. Castillo-Medina et al. [12,13,14,15] demonstrated typical features of SBiP1 as a chaperone that is inactive under darkness when it is phosphorylated on Thr and activated by Thr dephosphorylation upon the light stimulus [14,16]. This activation mechanism was shown to be highly responsive to light since SBiP1 Thr dephosphorylation occurred with as little as 1 µm photon m^−2^ s^−1^ [14], implicating a highly sensitive photoreceptor upstream of the activating phosphatase(s) [15]. In addition to light, heat but not cold stress modulated the chaperone activity by dephosphorylation. It was shown that heat stress stimulated SBiP1 to activate by Thr dephosphorylation even under darkness [14,15]. 

More recently, Castillo-Medina et al. [15] demonstrated a close link between nutrient metabolic cues and activation/inactivation of the chaperone by dephosphorylation. They found that inhibition of protein synthesis by cycloheximide also inhibited the chaperone activation, as it remained phosphorylated even after the light stimulus. Analogously but not identically, inhibition of glutamine biosynthesis by glufosinate, a competitive inhibitor of GS-GOGAT, delayed but did not completely prevent the light-induced stimulation of the chaperone [15]. Interestingly, it was recently reported that the cnidarian hosts regulate symbiont growth by limiting their carbon and nitrogen availability [36], suggesting that SBiP1 may be a highly relevant molecule in the regulation of these pathways in cnidarian-dinoflagellate relationships [15]. These data indicated that BiP-like proteins are crucial and stress-, light-, and nutrient-regulated molecular chaperones responsive to highly sensitive phototransduction mechanisms in Symbiodiniaceae.

## 3. Structural Insights of BiP Proteins

Among the diverse array of molecular chaperones, BiP proteins act as critical regulators of cellular homeostasis, maintaining and facilitating proper protein folding and translocation of nascent polypeptides [11]. BiPs also participate in stress responses via the UPR [9]. In the context of Symbiodiniaceae, the examination of this protein family is of exceptional significance due to their pivotal roles in the adaptation of these organisms to various environmental stressors, including fluctuations in temperature and light, which are characteristic of their habitat [20]. Therefore, we performed a comprehensive analysis of the conserved motifs and domain architecture of BiP proteins within Symbiodiniaceae, as well as modeling of three-dimensional (3D) structures with AlphaFold v1.5.5 colab software [37]. This helps not only to understand their functional intricacies but also to infer if the underlying mechanisms that facilitate the adaptation and survival of these organisms in their dynamic environments are conserved within their phylum.

Hsp70/BiP chaperones include both cytosolic and ER members. Both types are composed of two functional major domains, the N-terminal nucleotide binding domain (NBD) and the C-terminal substrate-binding domain (SBD), both joined by a flexible conserved linker. The main difference between them is that ER BiP proteins have a C-terminal HDEL or KDEL motif for ER retention while cytosolic proteins have an EEVD C-terminal motif; thus, SBiP1 presented the HDEL motif (Figure 1 and Figure 2B, black boxed and C-terminal motif, respectively), which was observed as a C-terminal tail in the tertiary structure (Figure 2A and Figure 3, red tails). The NBD is an ATPase domain composed of four subdomains named IA, IB, IIA, and IIB (Figure 3, inset), arranged in two lobes separated by a hydrophobic cleft; this domain is responsible for the hydrolysis of ATP—at the bottom of the cleft—which provides the energy required for protein folding and client release. The SBD includes two subdomains, an alpha-helical SBDα and a two-layered twisted β-sandwich SBDβ (Figure 3); this domain recognizes and binds to exposed hydrophobic regions of proteins that are in the process of folding [10,11]. Additionally, all BiP proteins have three distinctive groups of amino acid residues known as Hsp70 signatures: IDLGTTyS, DLGGGTfD, and IvLvGGsTRIPkIqK (Figure 2) [38,39].

In terms of conformational states, BiP proteins can exist as ATP- or ADP-bound. When ADP is bound, also known as apo or closed conformation, BiP has high affinity for substrates with a low dissociation rate because the SBDα subdomain acts as a lid for the SBDβ subdomain so that the hydrophobic residues from the substrate get trapped by the positively charged hydrophobic core of the SBDβ subdomain. Moreover, there is little interaction between the two major domains (NBD and SBD). When ATP is bound (also known as open conformation)*,* both NBD and SBD domains couple, and the dissociation rate increases, thus diminishing the affinity of BiP towards its substrates. ATP binding induces the rotation of the NBD lobe I relative to lobe II that results in the closure of the nucleotide binding cleft and the opening of the substrate cavity. This rotation also displaces the conserved Lys70 and Glu171 (DnaK, *E. coli*) by about 2 Å. These residues are proposed to stabilize the transition of the ATP-phosphate, while Glu171 together with Thr199 favor the inline nucleophilic attack on it [10,11]. Thus, BiP-assisted protein folding depends on the ATP/ADP exchange for substrate binding and release. This nucleotide exchange is mediated by nucleotide exchange factors (NEF) and co-chaperones such as J-domain proteins (JDP). NEF regulates the exchange from ADP to ATP, while JDP stimulates the ATPase activity by binding to the substrates or close to their cellular locations [11]. 

Because both the sequence and structure of Hsp70 family proteins are highly conserved, we took advantage of these traits to look for Hsp70-like proteins in members of the Symbiodiniaceae family through a series of Blastp, Blastn, tBlastn, and Blastx alignments using the nucleotide or protein sequence of SBiP1 of *S. microadriaticum* CassKB8, which belongs to the endoplasmic reticulum (ER Hsp70 subtype). In our analyses, we utilized amino acid, genome, and transcriptome databases, specifically focusing on the Symbiodiniaceae family. While the different blast searches can recover numerous sequences with high identity, we applied stringent criteria to ensure the relevance and accuracy of our dataset. We excluded sequences that were duplications, sequences from uncultured organisms, and redundant sequences from species data. This approach allowed us to focus on the most pertinent sequences, thereby enhancing the reliability of our analysis. We found SBiP1 homolog sequences in other 12 members of the Symbiodiniaceae family: *Breviolum* (1 sequence), *Cladocopium* (2 sequences), *Durusdinium* (1 sequence), and *Symbiodinium* (8 sequences) (Table 1). In addition, a multiple amino acid sequence alignment of the fourteen Symbiodiniaceae sequences and cytosolic and ER BiP protein sequences from plants, animals, and protists (*Arabidopsis thaliana, Chlamydomonas reinhardtii, Danio rerio, Drosophila melanogaster, Mus musculus, Plasmodium falciparum*) showed the three Hsp70 signatures and the C-terminal conserved motif for ER localization among them (Figure 2B). The sequences missing the signatures were only those too incomplete to span the conserved motifs. Finally, we modeled the 3D structures of Symbiodiniaceae BiP proteins except the ones from *D. trenchii* and *S. kagawutti* because they were incomplete; on the other hand, *Cladocopium* sp. C3 was also reported as a partial sequence, but it was possible to distinguish the SBD domain and a partial NBD domain. We observed that Symbiodiniaceae BiPs were conserved not only at the BiP primary amino acid sequence level but also at the tertiary structural level (Figure 4). This indicates a high conservation of sites for interaction with other ligands and for activity, function, and regulation.

## 4. Regulation of *BiP* Gene Expression

Originally, the Symbiodiniaceae family of marine dinoflagellates with fifteen extant and divergent lineages was grouped into only one genus, *Symbiodinium* [40,41]. It was later reported that each evolutionarily divergent *Symbiodinium* clade could be grouped into diverse genera within the Symbiodiniaceae family, and, thus, it is now recognized that they comprise different genera with a broad genetic divergence among them [40]. For instance, comparing the sequence genomes of *S. microadriaticum*, *S. minutum*, and *S. kawagutii* showed that the overall genome organization included differences in gene orientation and content with specific attributes for each genus [40]. 

For the last decade, an increasing number of scientific works have been developed attempting to understand the molecular mechanisms underlying the symbiosis between Symbiodiniaceae and cnidarians, generating vast amounts of genomic and transcriptomic data [42,43]. The Hsp proteins and genes have been studied regarding their involvement in the coral-dinoflagellate symbiosis and, more precisely, their role in the thermal stress response [22,44,45,46].

As mentioned before, the expression of *Hsp* coding genes has been mostly studied in the context of the Symbiodiniaceae-cnidarian symbiosis during thermal stress, as they are molecular chaperones involved in the heat stress response, and elevated seawater temperature due to global warming is a current threat to coral reefs. Over the years, more information about the role of the Hsp proteins in the cellular response of Symbiodiniaceae to elevated temperatures became available thanks to the advent of the “omics” technologies; unfortunately, the few studies that revealed changes in expression of proteins of the Hsp family were not BiP protein chaperones. Furthermore, although these omics approaches could provide insight into the *Hsp* gene expression patterns under thermal stress, they failed to explain the molecular mechanisms governing them. For example, there are *cis*-regulatory elements (CRE) that may explain the up- and down-regulation of the *Hsp* gene expression, derived from reports that suggested that the differences in the promoter regions may explain the different expression of BiP-coding genes [47]. Additionally, in plants, the *Hsp* coding genes are transcriptionally regulated by heat shock factors (HSF) that bind to discrete palindromic sites (heat shock elements; HSE) located within the promoters [48]. An analysis of *Hsp* gene families in barley identified 30 relevant motifs located in 1-kb upstream regulatory regions. The more relevant were the CBF-binding site, CCAAT box, ASF-1-binding site, and ABRE-like core motifs of DRE/CRT, which are related to the dehydration response, and regulation of heat shock related responses [49]. The CCAAT box exists in essentially all *Hsp70* promoters and maintains an open chromatin configuration so that the HSF can bind rapidly to the promoter [48]. The CBF and DRE/CRT motifs (C-repeat and dehydration response element) are present in cold-regulated promoters but are also involved in different responses to stress, including heat [50]. In *Arabidopsis thaliana* and *Solanum tuberosum cis*-regulatory elements belonging to 30 families of transcription factors have been reported for *BiP* promoters. They are involved in cell differentiation, organ development, hormonal responses, and metabolic processes in response to temperature and water deficit stress [47]. These investigations show the importance of studying the promoter architecture of *Hsp* genes to understand their regulation.

For Symbiodiniaceae, to our understanding, there is no information related to the *Hsp* promoter analysis, although the importance of such analysis has been widely demonstrated. There is, however, one study reporting that all Symbiodiniaceae (then referred to as a single *Symbiodinium* genus) genera promoter elements are located within 1 kb of the start codon, and the polyadenylation signal is found about 300 nucleotides after the stop codon [51]. Thus, we analyzed the promoters of the seven *BiP* coding sequences (Table 1) with the software New PLACE version 30.0 that predicts plant promoters [52], focusing on response to stress related elements. The available DNA sequences contained between 700 and 900 nucleotides upstream the start codon, except for *S. kawagutii* CCMP2468, which had only 79 annotated nucleotides upstream the start codon. This limited length may account for the lack of CRE detection in the *BiP* genomic sequence of this species. Indeed, all but *S. kawagutti* CCMP2468 contained the consensus CCAAT box found in promoters of heat shock genes as reported for *Arabidopsis* [53]. We also found the consensus sequence of the plastid response element (PRE) reported as an enhancer at the promoter of *Chlamydomonas Hsp70* [54] in all the sequences except for *S. natans* and *S. kawagutti* CCMP2468. Interestingly, only one UPR element, Motif I from *A. thaliana Hsp90* [55], was found and present only within the promoter of *S. pilosum BiP5.* Other notable CRE included CBF and DRE motifs related to low temperature and dehydration. It is noteworthy that even though we failed to find more stress-related regulatory elements, these results must be interpreted carefully since they were derived from a plant database and the binding sequences for those heat shock elements may be different in Symbiodiniaceae. The complete set of CRE found is available at Appendix A.

## 5. BiP Protein Regulation and Expression Patterns in Dinoflagellates

A BLAST analysis with the SBiP1 nucleotide, protein, and translated nucleotide as queries against Dinoflagellata (taxid: 2864) yielded numerous sequences with high identity. However, we limited our analysis to only those sequences presenting the HDEL motifs at their C-termini and having more than 63% identity (Table 2). The scarce number of sequences reported in the NCBI for dinoflagellate BiPs indicates how poorly characterized this protein is in this phylum. Nevertheless, the non-photosynthetic dinoflagellate *Crypthecodinium cohnii* BiP sequence has been useful for intra-alveolate phylogeny analyses and supports the existing relationship within this group (the major alveolate groups are ciliates, apicomplexans, and dinoflagellates). By phylogenies of individual genes including the ciliates, apicomplexan, and *C. cohnii* genes (*BiP*, *hsp70*, *hsp90*, *hspl0*, *actin*, and *β-tubulin*), it was concluded that apicomplexans and dinoflagellates are sister groups because they branch together and exclude the ciliates, although not all gene trees support this conclusion [56].

In terms of *BiP* gene expression in other dinoflagellates, several interesting features have also been observed. When the transcriptome in the toxin-producing dinoflagellate *Alexandrium catenella* was analyzed through the paralytic shellfish toxin biosynthesis stages within the cell cycle, the *BiP* gene was one of the ten genes that changed its expression profiles. This suggested that protein synthesis could be synergic to toxin biosynthesis, where proteins involved, such as BiP, may play an important role [57]. Additionally, a proteome analysis of light regulation in Symbiodiniaceae clades B and D showed that most of the proteins differentially regulated by light were involved in protein folding, sorting, and degradation. It could then be implicit that BiP proteins usually respond to light stimuli in these microorganisms and at different regulation levels. For example, BiPs are known to be up-regulated at low and high light intensities [58]. In addition, Castillo-Medina et al. [12,13,14,15] discovered that the BiP homolog from *S. microadriaticum* CassKB8, SBiP1 responded to heat stress, nutritional cues, and light stimuli by a regulatory mechanism at the posttranslational level (without changes in BiP protein expression). Of particular interest was the light-stimulated response: SBiP1 phosphorylated in threonine under darkness and undergoing dephosphorylation upon exposure to light—a behavior likely related to circadian rhythms. Circadian rhythms are endogenous clocks—existing in most organisms—that respond to environmental rhythmic cues, such as light and temperature. In *Symbiodinium* spp., photosynthesis possesses a diel rhythmicity under light/dark cycles both as a free-living organism and during symbiosis [59]. Examples of such circadian rhythms exist in other dinoflagellates, including *Lingolidonium polyedra,* where photosynthesis shows a diurnal rhythm [60], and *Pyrocystis lunula*, where bioluminescence (scintillions and plastid ultrastructure) is related to day and night phases [61]. However, regarding the heat shock proteins, apart from SBiP1, only in the coral host *Acropora pruinosa* an *Hsp70* coding gene was seen up-regulated during daytime [62]. Given that both the structure and the sequence of Hsp/BiP in different species is quite conserved, it is possible that a similar regulation could occur in Symbiodiniaceae. Despite the scarce information available on BiP proteins in dinoflagellates, these data suggest that these proteins play roles involved in the mechanisms of toxin biosynthesis and in the cellular response to light.

## 6. Phylogenetic Analysis Shows Conservation and Clustering of Dinoflagellate BiP Proteins

Phylogenetic analyses have shown that BiP proteins are highly conserved across different species, with the highest sequence identity found between homologs from closely related species [63,64,65]. For Dinoflagellata, it has been demonstrated that Hsp70 clusters in one clade belonging to Alveolata along with organisms of the classes Apicomplexa and Perkinsea [66]. We performed a phylogenetic analysis (Figure 5) that clearly clustered the Hsp70 sequences analyzed into two main groups: one comprising Symbiodiniaceae and other dinoflagellates (*C. cohnii*, *Amoebophyra*, and *Perkinsus*), as well as other protists of the Alveolata clade (apicomplexans such as *Cryptosporidium, Plasmodium, Eimeria,* and *Toxoplasma*); the other group clustered plant and animal Hsp sequences. Interestingly, the latter also grouped with the ciliophores *Blepharisma* and *Pseudourostyla,* which also belong to the Alveolata. These results indicate that BiP/Hsp sequences are conserved within species, most likely also reflecting functional conservation.

## 7. Conclusions and Future Directions

Considering the relevance of Symbiodiniaceae-cnidarian symbiosis as the foundation of the coral reef ecosystems and that of BiP chaperone function in the mechanisms underlying the stressful events that lead to the disruption of the symbiosis, the little information available reveals an important gap of information and research regarding the molecular mechanisms governing both the expression and function of these proteins in Symbiodiniaceae and even in dinoflagellates in general. Our work reveals a high degree of conservation in both Symbiodiniaceae and the present available dinoflagellate BiP sequences at the gene and protein levels as well as at the three-dimensional structural one. This is also a result of the close relationship found among the dinoflagellates, which suggests evolutionarily conserved function and regulation mechanisms, a notion that will, of course, be subject to experimental demonstration. A possible link between circadian clocks and BiP regulation should also be considered. Finally, this review highlights the need to carry out substantial experimental work to fully understand the function and regulation of these proteins in the physiology of both free-living Symbiodiniaceae and the various endosymbiont stages. Investigating gene regulation, promoter architecture, and three-dimensional structural conservation will provide critical insights into the various BiP contributions to the functionality of Symbiodiniaceae and resilience of their hosts. These multifaceted approaches will illuminate the intricate dynamics of heat shock proteins resident of the ER involved in cell survival and will also pave the way for a deeper understanding of the delicate balance that sustains coral reef ecosystems.

## Figures and Tables

**Figure 1 microorganisms-12-02126-f001:**
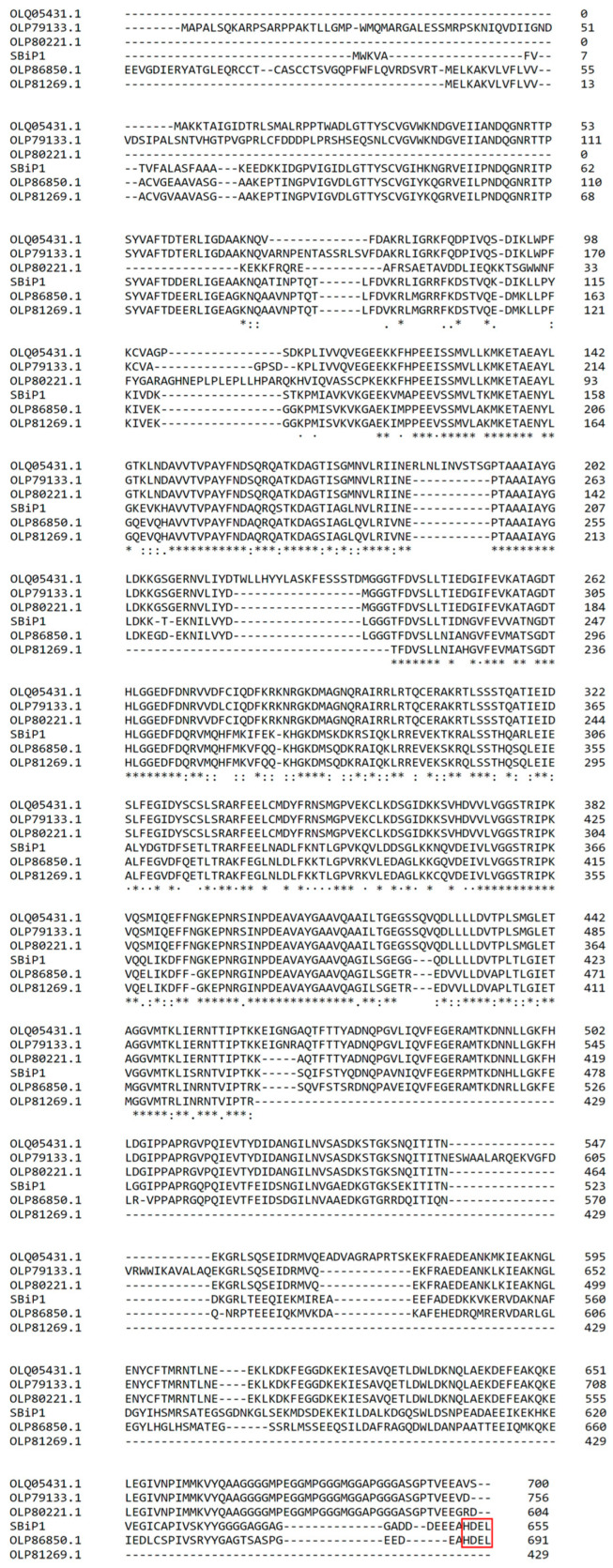
Multiple Alignment of the *S. microadriaticum* CassKB8 SBiP1 sequence (SmicHSP75) with five other annotated homologous Hsp70 family protein sequences from *S. microadriaticum* [34] with accession numbers OLP86850.1, OLP81269.1, OLP79133.1, OLQ05431.1, and OLP80221.1. Sequences with accession numbers OLP86850.1 and OLP81269.1 were annotated also as BiP-like proteins. The SBiP1 sequence is annotated in the GenBank as OP429595.1 [14]. The boxed sequence (in red) at the C-terminal shows the ER localization sequence in SBiP1 and OLP86850.1. Asterisks indicate identical amino acid positions for all sequences; colons indicate different amino acids with similar chemical properties; and periods indicate weakly conserved amino acids.

**Figure 2 microorganisms-12-02126-f002:**
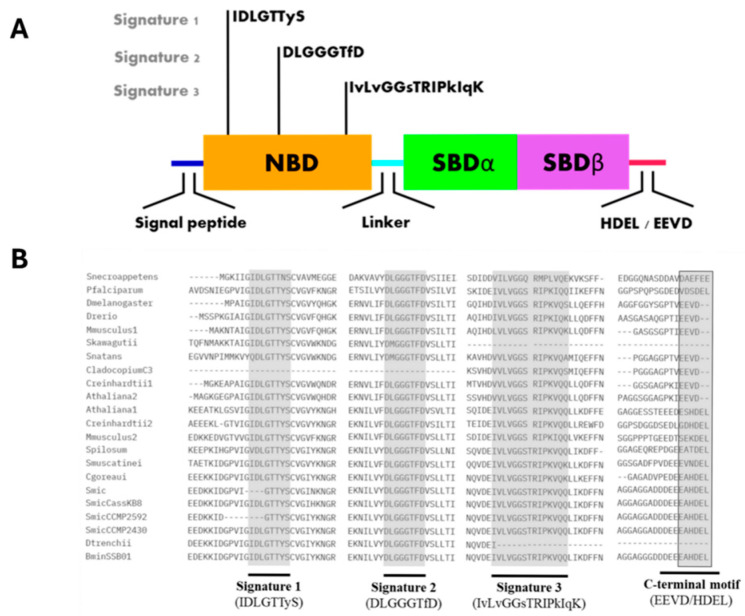
Domain architecture of SBiP1 (**A**) and Hsp70 signatures of BiP homologs (**B**). The characteristic three amino acid signatures of Hsp70 proteins are found within the major SBiP1 nucleotide binding domain (**A**; NBD, orange). The amino acid sequences of the signatures are conserved among different species as well as among members of the Symbiodiniaceae family (**B**). The C-terminal end (**B**; black boxed sequences) differentiates between ER-resident BiP proteins, which contain the HDEL endoplasmic reticulum retention motif, and cytosolic Hsp70 proteins, which have the EEVD motif instead. The nucleotide binding domain (NBD) is shown in orange, and the two subdomains of the substrate-binding domain (SBD) in green (α) and magenta (β).

**Figure 3 microorganisms-12-02126-f003:**
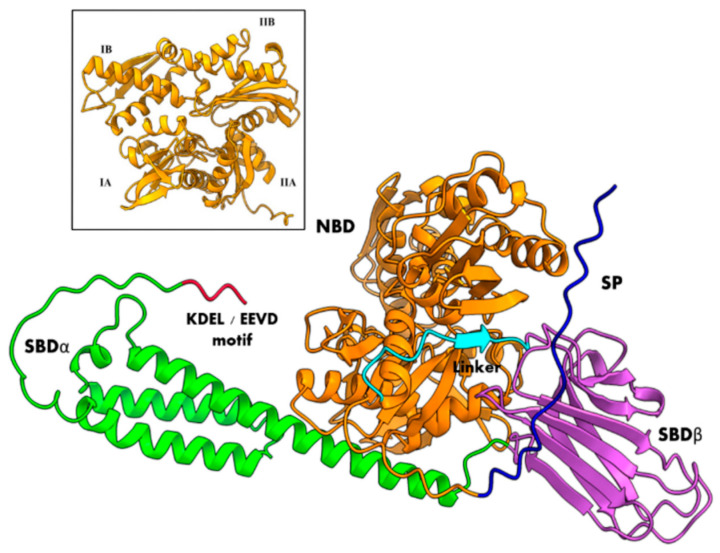
Representation of the canonical structure of Hsp70/BiP chaperones from the SBiP1 amino acid sequence. The three-dimensional model shows the N-terminal nucleotide binding domain (NBD, orange) with its four subdomains (inset) and the C-terminal substrate-binding domain (SBD) with its two subdomains: alpha in green and beta in lilac. The NBD and SBD are joined by a flexible conserved linker (light blue). Also visible are the signal peptide (dark blue) and the C-terminal cytosolic or endoplasmic reticulum retention motif (red).

**Figure 4 microorganisms-12-02126-f004:**
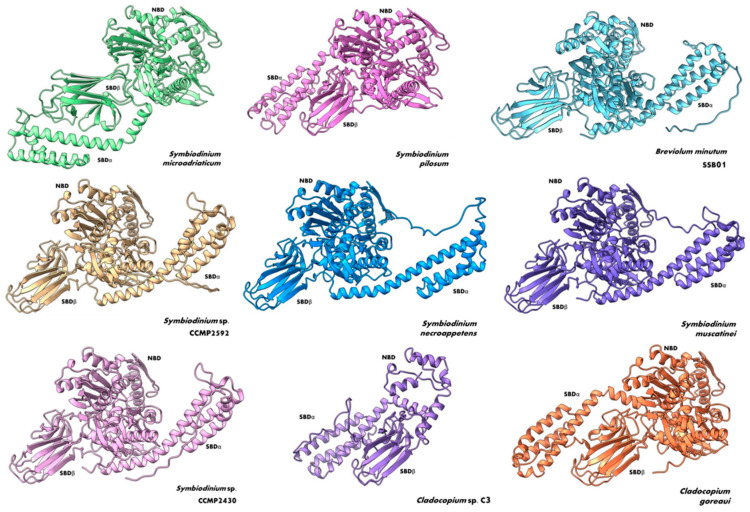
Three-dimensional structures of Symbiodiniaceae BiP proteins. Three-dimensional models of BiP protein sequences in members of the Symbiodiniaceae family. For each model, the two major subdomains, the nucleotide-binding domain (NBD) and the two modules, alpha (SBDα) and beta (SBDβ), of the substrate-binding domain are indicated.

**Figure 5 microorganisms-12-02126-f005:**
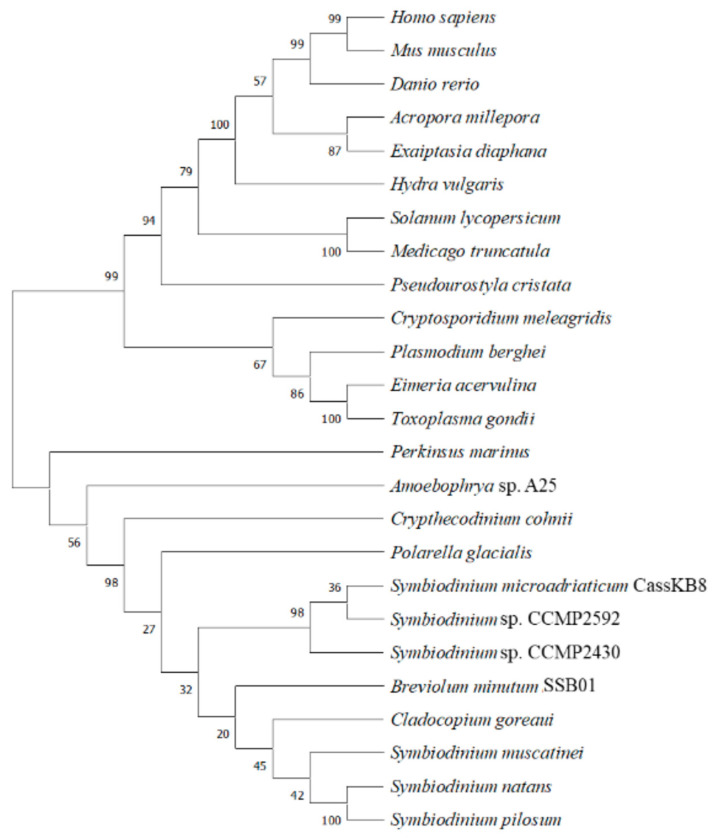
Maximum likelihood phylogenetic tree of BiP protein sequences from Symbiodiniaceae, other dinoflagellates, and plants and animals. Two major groups are observed: one (**bottom**) comprising the Symbiodiniaceae members, and another (**top**), which encompasses sequences from protists, plants, and animals. Numbers indicate the percentage of trees in which the associated taxa clustered together.

**Table 1 microorganisms-12-02126-t001:** SBiP1 homologous sequences in the Symbiodiniaceae family. Listed are the names of the organisms where they belong, the type of sequence, their subcellular localization, and accession numbers. The details of their *cis*-regulatory elements found are also included. NP: Analysis was not performed because the sequence was too short or nonexistent.

Organism	Name	Sequence Type	Subcellular Localization	*Cis*-Regulatory Elements	Accession No.
*Symbiodinium* *microadriaticum* Cass KB8	SBiP1	Protein	ER	CCAAT-box, PRE, CBF/DRE	OP429595
*Symbiodinium* sp. CCMP2592	BiP5	Protein	ER	CCAAT-box, PRE, CBF/DRE	CAE7227403
*Symbiodinium* *microadriaticum*	BiP5	Protein	ER	CCAAT-box, PRE, CBF/DRE	OLP91134
*Symbiodinium pilosum*	BiP5	Protein	ER	CCAAT-box, PRE, CBF/DRE, UPR-I	CAE7221339
*Symbiodinium natans*	-	Protein	ER	CCAAT-box, CBF/DRE	CAE7360486
*Symbiodinium necroappetens*	carB	Protein	Cytosol	CCAAT-box, CBF/DRE	CAE7932356
*Cladocopium* sp. C3	Hsp70	Protein	Cytosol	NP	ABA28988
*Symbiodinium* sp. CCMP2430	*Hsp70*	mRNA	ER	NP	HBTH01040005
*Breviolum* *minutum* SSB01	-	mRNA	ER	NP	GICE01029930
*Cladocopium goreaui*	-	mRNA	ER	NP	ICPI01002597
*Durusdinium trenchii*	-	mRNA	NP	NP	ICPJ01013860
*Symbiodinium muscatinei*	-	mRNA	ER	NP	GFDR03033717
*Symbiodinium kawagutii* CCMP2468	-	mRNA	NP	NP	KC950716

**Table 2 microorganisms-12-02126-t002:** SBiP1 homologous sequences in Dinoflagellata. Listed are the names of the organisms where each one belongs, the type of sequence, their subcellular localization, and corresponding accession numbers.

Organism	Sequence Name	Subcellular Localization	Accession No.
*Crypthecodinium cohnii*	*BiP*	ER	AF421538
*Prorocentrum minimum*	*Hsp70*	Cytosol	JN401970
*Akashiwo sanguinea*	*Hsp70*	cytosol	KJ755185
*Cochlodinium polykrikoides*	*Hsp70*	Cytosol	KP010830
*Heterocapsa triquetra*	*Hsp70*	Unknown	AY729868
*Polarella glacialis*	Unnamed	ER	CAE8584766
*Amoebophyra* sp. A25	Unnamed	ER	CAD7976323
Uncultured dinoflagellate	*Unnamed*	Cytosol	GU555406

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
