# Peer review of "BiP Proteins from Symbiodiniaceae: A “Shocking” Story"

_microorganisms, 2024, doi:10.3390/microorganisms12112126_

Round 1

Reviewer 1 Report

Comments and Suggestions for Authors

The manuscript (MS) by Estefanía Morales-Ruiz et al. reviewed the BiP proteins in the family of the Symbiodiniaceae. The topic of the MS seems sound and interesting for the audiences. However, the MS writing still need important improvement before its acceptance.

1. Abstract, its too general. Please consider to add some comprehensive and in-depth discussion for the BiP proteins.

2. Bip, or BIP? Which one is correct? or both?

3. The subtitle names, suggest to improve them with more information, the current names are too short to be fully understood.

4. The name, Symbiodiniaceae is for a family in the taxonomy, consider to list it as italic.

5. Why some words were list as the bold form, such as BiP in Line 29, ER in Line 45?

6. sp., is not italic, table 1. also, the format of this table need corrected. Also for table 2.

7. Line 13, "folding during translation was documented" may have a syntax error, it is suggested to replace "was" with "were".

8. Line 87, “the presence of Hsp70 and Hsp90 chaperones is considered”, suggest changing “is” to “are”.

9. Line 106, “is still far from understood”, suggest changing “is” to “are”.

10. Figure 1, 3, it is recommended to replace it with a clearer photo.

11. Line 197, “named IA, IB, IIA, IIB” add “(” before “IA”.

12. Line 341, delete “)”.

13.  All figures in the MS are not clear enough, they must be updated, and provide a better one with higher resolution.

14. For the reference, they're some errors found, please check and correct them.

Comments on the Quality of English Language

The language of the MS is fine.

Author Response

All modifications regarding comments from this reviewer are highlighted in yellow in the revised manuscript.

Comment 1. Abstract, it’s too general. Please consider to add some comprehensive and in-depth discussion for the BiP proteins.

Response 1. The abstract has been modified to include: a) enough background on BiPs; b) justification for reviewing BiP proteins in Symbiodiniaceae and dinoflagellates; c) what is lacking in the field on BiP research; and d) the overall findings when searching the literature and carrying out our analyses; and e) we finish with the conclusion of our review on the subject. We believe this has made the abstract more concise and we thank the reviewer for the request. We considered though, that comprehensive and in-depth discussion on BiPs would make it a lengthy abstract, and that such material is more appropriate within the body of the paper.

Comment 2. Bip, or BIP? Which one is correct? or both?

Response 2. The convention is to name them as “BiP”. However, BIP was used in table 1 for BIP5 because it was listed as such in the data base. We renamed them as BiP5 for uniformity.

Comment 3. The subtitle names, suggest to improve them with more information, the current names are too short to be fully understood.

Response 3. We have added more information to the subtitles as follows.

“1. Introduction” to “1. Introduction: Discovery and Functional Insights of BiP Proteins”

“2. BiP proteins in Symbiodiniaceae” to “2. BiP Proteins in Symbiodiniaceae: Discovery, Stress Responses, Light-Regulated Phosphorylation and Possible Roles”

“3. BiP” protein structure to “3. Structural Insights of BiP Proteins”

“4. Gene expression” to “4. Regulation of BiP Gene Expression” 

“5. BiP proteins in dinoflagellates” to “5. BiP Protein Regulation and Expression Patterns in Dinoflagellates”

“6. BiP phylogeny” to “6. Phylogenetic Analysis Shows Conservation and Clustering of Dinoflagellate BiP Proteins” 

“7. Conclusion” to “7. Conclusion: Conservation, Functional Implications of BiP Proteins in Symbiodiniaceae and Future Research Directions”

Comment 4. The name, Symbiodiniaceae is for a family in the taxonomy, consider to list it as italic.

Response 4. Symbiodiniaceae as family is written without italics. Further confirmed after consulting the following sources:

ScienceDirect (https://www.sciencedirect.com/topics/agricultural-and-biological-sciences/symbiodiniaceae)

Algaebase.org (https://www.algaebase.org/browse/taxonomy/#167698)

Symbiodiniaceae style guide (https://www.thelifeaquatic.net/?page_id=292)

Wikipedia (https://en.wikipedia.org/wiki/Symbiodiniaceae)

Therefore, we left the family name unchanged.

Comment 5. Why some words were list as the bold form, such as BiP in Line 29, ER in Line 45?

Response 5. It was done to emphasize the term but it seems to add noise to the writing style so we removed the boldface on the text.

Comment 6. sp., is not italic, table 1. also, the format of this table need corrected. Also for table 2.

Response 6. We corrected the italicized “sp.” to normal text. Table formatting will be done by the journal when in production if the manuscript is accepted.

Comment 7. Line 13, "folding during translation was documented" may have a syntax error, it is suggested to replace "was" with "were".

Response 7. With the changes on the abstract text, this no longer applies.

Comment 8. Line 87, “the presence of Hsp70 and Hsp90 chaperones is considered”, suggest changing “is” to “are”.

Response 8. “is” is correct since its noun is “the presence”; no change was made.

Comment 9. Line 106, “is still far from understood”, suggest changing “is” to “are”. 

Response 9. Thank you, our mistake and change made accordingly (line 115).

Comment 10. Figure 1, 3, it is recommended to replace it with a clearer photo.

Response 10.. The figures were improved.

Comment 11. Line 197, “named IA, IB, IIA, IIB” add “(” before “IA”.

Response 11. Thank you for the observation, it was our mistake; actually, the “)” was extra and was removed (line 208).

Comment 12. Line 341, delete “)”.

Response 12. Thank you, it was removed, and this led to another correction; “query” changed to “queries” (line 375).

Comment 13. All figures in the MS are not clear enough, they must be updated, and provide a better one with higher resolution.

Response 13.. The resolution of figures was improved.

Comment 14. For the reference, they're some errors found, please check and correct them.

Response 14. Mistakes have been corrected.

Reviewer 2 Report

Comments and Suggestions for Authors

The title is intriguing, but at first I was not sure why the story of BiP proteins was shocking. Coming back to it a few hours later, I realised it may be a reference to heat shock? If it is only to make an interesting title, maybe put shocking in quotation marks? Alternatively, are there any aspects of the BiPs or their functions that really are shocking in the sense of unexpected?

My major issue with this review is that the transcription of the heat shock genes seems poorly described. In particular, the authors describe what appear to be transcriptional cis regulatory sequence motifs in the 5’UTR. My view is that transcriptional regulatory sequences are unlikely to be found in this (transcribed) region of the gene. Analysis of transcriptional regulatory sequences in dinoflagellates is also problematic as, due to the presence of trans splicing, the transcriptional start site cannot be determined from the mature RNA sequence. I also did not find any discussion of relict spice leader sequences which could contribute a TTTT motif upstream of the start codon for some genes. These issues must be adequately addressed before the MS can be accepted for publication.

I had a few questions concerning Table 1:

TATA box should not be used, perhaps TATA-like box?

Why is there a TTTT box just before the start codon? Would this not place it in the 5’UTR? How long is the 5’UTR in these sequences? If regulatory elements are being examined in the 5’UTR, why not include sequences from the many transcriptomes that are available? The authors describe analysing up to 50 nucleotides upstream from the start codon, but if this places the sequences within the 5’UTR they are unlikely to be transcriptional control elements.

Some dinoflagellate sequences may contain a relict spliced leader sequence that has been returned to the DNA sequence from a trans spliced mRNA. This sequence, 5’-DCCGUAGCCAUUUUGGCUCAAG-3’, would contain a TTTT sequence once returned to the DNA. Why do the authors think that this would now become a regulatory sequence? Are there any other parts of the SL that appear in the sequence?

OP429585 returns a cytochrome c from Tantilla sp, no idea what this is. Also, the sequences marked as protein begin at the start codon and do not contain any upstream sequences.

Table 1 also references a CCAAT-box, which I presume has a cognate binding protein in other systems. The dinoflagellates generally have a reduced complement of DNA binding proteins, so have the authors checked they do in fact have a binding protein that can bind CCAAT?

As a general rule, the TTTT motif should be a transcriptional cis regulatory sequence binding a TBP-like factor (TLF), upstream from the transcriptional start site. However, I assume that these dinoflagellate sequences are modified by an addition of an spliced leader making the authentic transcriptional start site difficult to pinpoint. How are the authors able to identify which TTTT sequences are potential regulatory elements? How do they identify other regulatory sequences, and where are they in the gene sequence?

Line 99 Hsps do not have a symbiotic relationship, the dinoflagellates do.

Line 247 How were the 3D structures modeled?

Line 275 The conservation of promoter sequences between species depends on accurate identification of regulatory sequences in individual species (see above).

Line 284 How much of the 50 nucleotides upstream from the start codon is likely to be a promoter sequence and how much 5’UTR? This can be evaluated from the transcript sequences.

Line 288 what does C<G mean?

Line 299 What insights about Hsp expression in response to thermal stress have been revealed? It certainly seems relevant to discuss this aspect. Are there systematic changes in RNA or protein levels in addition to the changes in phosphorylation?

Line 325 Was the 1 kb upstream region of all genes used as a putative promoter? Or thethe 50 nucleotides mentioned in line 284?

Line 334 How was the length of the S kawagutii promoter determined to be 79 nucleotides? Is there another gene 79 nucleotides upstream of the start codon?

Line 341 remove bracket.

Line 342 What database was used for this recovery of only 8 sequences? A search comparing SBiP1 protein sequence with the NCBI TSA database (taxid 2864) using TBLASTN recovers at least a hundred sequences with >86% identity.

Line 347 Some caution should be exercised when discussing gene expression. There is not necessarily any relation between changes in RNA levels and changes in protein levels.

Line 368 A light response may also be just a response to light rather than a circadian behavior.

Line 373 Check spelling of dinoflagellate used.

Author Response

All changes with addressed issues by this reviewer are highlighted in grey in the revised manuscript.

Comment 1. The title is intriguing, but at first I was not sure why the story of BiP proteins was shocking. Coming back to it a few hours later, I realised it may be a reference to heat shock? If it is only to make an interesting title, maybe put shocking in quotation marks? Alternatively, are there any aspects of the BiPs or their functions that really are shocking in the sense of unexpected?

Response 1. The intent of the title is indeed due to the “heat shock” nature of the expression induction of the proteins so the “shocking” in the title was quoted.

Comment 2. My major issue with this review is that the transcription of the heat shock genes seems poorly described. In particular, the authors describe what appear to be transcriptional cis regulatory sequence motifs in the 5’UTR. My view is that transcriptional regulatory sequences are unlikely to be found in this (transcribed) region of the gene. Analysis of transcriptional regulatory sequences in dinoflagellates is also problematic as, due to the presence of trans splicing, the transcriptional start site cannot be determined from the mature RNA sequence.

Response 2. Thank you for your feedback. We believe our way to describe the transcriptional element analysis was unclear. We did analyze the genomic sequences and not the 5’ UTR’s from the transcripts so we have indeed analyzed the regions containing the promotors and cis-regulatory sequences. Transcriptional regulatory elements, such as cis-regulatory elements (CREs), are typically not found in the transcribed region of the gene; namely, the 5'UTR. Instead, these elements are generally located in the upstream, non-transcribed regions of the DNA sequence, particularly in the promoter and enhancer regions. In Symbiodiniaceae however, a large part of the genomic sequence with some regulatory elements close to the ATG initiation codon is conserved in the transcript even after splicing; this may have given the wrong idea that we are analyzing the transcript itself for regulatory elements, but this was not the case. To make our point more clearly, we have revised the text to avoid confusion in this respect (lines 312-316).

Comment 3. I also did not find any discussion of relict splice leader sequences which could contribute a TTTT motif upstream of the start codon for some genes.

Response 3. It has been reported that the UUUU sequence in the SL-RNA is proposed to constitute the binding site for Sm proteins required for the trans-splicing. After splicing, this nucleotide stretch is irrelevant for transcription once the mature transcript is generated. Since we did not analyze transcripts for the presence of gene regulatory elements for transcription, further discussion on the subject is unnecessary.

Comment 4. I had a few questions concerning Table 1:

TATA box should not be used, perhaps TATA-like box?

Response 1. TATA box was substituted by TATA-like box.

Comment 5. Why is there a TTTT box just before the start codon? Would this not place it in the 5’UTR?

Response 5. The TTTT motifs we found on gene sequences were located between 16 to 24 nucleotides upstream the start codon. As mentioned in the manuscript, in dinoflagellates this TTTT motif replaces de canonical TATA-box which is usually located at a similar distance from the start codon in DNA sequences and constitutes the site for transcription initiation. Again, we did not analyze the 5´UTR from the transcripts to search for regulatory elements. The analysis was carried out on the corresponding genes. The addition of the trans-spliced leader sequence (SL) to the 5' end of the pre-mRNA occurs post-transcriptionally, meaning it does not influence the transcription start site or promoter activity. As mentioned above, the UUUU sequence could be wrongly conceptualized as a regulatory TTTT sequence on the transcript but that is not the case. Furthermore, any regulatory element for DNA transcription is irrelevant for the transcribed mature mRNA.

Comment 6. How long is the 5’UTR in these sequences? If regulatory elements are being examined in the 5’UTR, why not include sequences from the many transcriptomes that are available? The authors describe analysing up to 50 nucleotides upstream from the start codon, but if this places the sequences within the 5’UTR they are unlikely to be transcriptional control elements.

Response 6. Precisely our point; our bioinformatics approach involved performing a series of BLAST analyses, targeting amino acid, genome, and transcriptome databases to identify homologous BiP sequences in Symbiodiniaceae and other dinoflagellates. However, many of the hits obtained from transcriptome and amino acid databases lacked the corresponding full gene (DNA) sequences. As a result, several downstream analyses, such as promoter region identification, could not be performed comprehensively. Indeed, we emphasize in the manuscript that this limitation highlights the incomplete nature of current genomic data for dinoflagellates and underscores the need for more complete genome sequencing to fully explore the functional and regulatory aspects of BiP proteins.

Comment 7. Some dinoflagellate sequences may contain a relict spliced leader sequence that has been returned to the DNA sequence from a trans spliced mRNA. This sequence, 5’-DCCGUAGCCAUUUUGGCUCAAG-3’, would contain a TTTT sequence once returned to the DNA. Why do the authors think that this would now become a regulatory sequence? Are there any other parts of the SL that appear in the sequence?

Response 7. To add to the already answered points above, there is no evidence so far, that the spliced leader (SL) sequence is integrated back into the genomic DNA after being added to the mRNA post-transcriptionally. The SL sequence is added through trans-splicing and does not typically return to or become part of the DNA sequence. Therefore, the assumption that it could form a regulatory sequence in the genome is not supported by current knowledge. Additionally, we have not observed any parts of the SL sequence reappearing in the DNA. Therefore, we did not consider this point relevant enough to be discussed in the manuscript.

Comment 8. OP429585 returns a cytochrome c from Tantilla sp, no idea what this is. Also, the sequences marked as protein begin at the start codon and do not contain any upstream sequences.

Response 8. Thank you for the observation, the accession number on the table had a mistake; the accession number was corrected to OP429595 which is the one for Symbiodinium microadriaticum subsp. microadriaticum binding immunoglobulin protein 1 (bip1) mRNA.

Comment 9. Table 1 also references a CCAAT-box, which I presume has a cognate binding protein in other systems. The dinoflagellates generally have a reduced complement of DNA binding proteins, so have the authors checked they do in fact have a binding protein that can bind CCAAT?

Response 9. We did find homologous dinoflagellate sequences to yeast and human CBF which is one of the CCAAT-box interacting proteins but this information was out of the scope of the review and thus it was not included.

Comment 10. As a general rule, the TTTT motif should be a transcriptional cis regulatory sequence binding a TBP-like factor (TLF), upstream from the transcriptional start site. However, I assume that these dinoflagellate sequences are modified by an addition of an spliced leader making the authentic transcriptional start site difficult to pinpoint. How are the authors able to identify which TTTT sequences are potential regulatory elements? How do they identify other regulatory sequences, and where are they in the gene sequence?

Response 10. The identification of cis-regulatory elements, including the TTTT motif, was conducted on the DNA sequences. Thus, the spliced leader sequence is a post-transcriptional modification and does not affect the TTTT motif located in the promoter region of the gene. Therefore, the TTTT motif and other regulatory elements were identified based on their presence in the upstream DNA sequences, which are typically associated with transcriptional regulation. We used the PLACE software to predict these elements, focusing on stress-related regulatory sequences. The exact positions of these elements within the gene sequences are detailed in our supplementary materials.

Comment 11. Line 99 Hsps do not have a symbiotic relationship, the dinoflagellates do.

Response 11. The line was changed to “further similar adaptations in other symbiotic relationships” (line 97).

Comment 12. Line 247 How were the 3D structures modeled?

Response 12. Three-dimensional structures were modeled using the AlphaFold colab notebook. We have included this information and its reference in the text (line 196).

Comment 13. Line 275 The conservation of promoter sequences between species depends on accurate identification of regulatory sequences in individual species (see above).

Response 13. We modified the sentence (lines 305-306).

Comment 14. Line 284 How much of the 50 nucleotides upstream from the start codon is likely to be a promoter sequence and how much 5’UTR? This can be evaluated from the transcript sequences.

Response 14. Not with the current sequence information available. Our primary objective was to investigate the conservation of the TTTT motif within the promoter region of the genes we analyzed. Therefore, we focused on the 50 nucleotides upstream from the start codon. For the distinction between the promoter region and the resulting 5’-UTR region of the mature mRNA after the trans-splicing event accurate determination can only be achieved through experimental comparison of both the DNA sequence and the mRNA sequence. However, the available DNA sequences lack their corresponding mRNA transcript sequences, which precluded us from further analyzing the UTR region. Future studies incorporating both DNA and mRNA sequences will be necessary to delineate the exact boundaries of the promoter and 5’UTR regions.

Comment 15. Line 288 what does C<G mean?

Response 15. It means that the expected Cytosine was replaced by a Guanine. However, since it seems to be confusing, we changed “<” to “/” to make it more obvious (line 318).

Comment 16. Line 299 What insights about Hsp expression in response to thermal stress have been revealed? It certainly seems relevant to discuss this aspect. Are there systematic changes in RNA or protein levels in addition to the changes in phosphorylation?

Response 16. This was originally discussed in lines 119-130 of the manuscript.

Comment 17. Line 325 Was the 1 kb upstream region of all genes used as a putative promoter? Or the 50 nucleotides mentioned in line 284?

Response 17. The promoter region analyzed varied from 700 to 900 nucleotides in length, depending on the available sequence data. Only one gene, S. kawagutti CCMP2468, had a shorter reported upstream region of 79 nucleotides. We have included this information in the manuscript (lines 358-363).

Comment 18. Line 334 How was the length of the S kawagutii promoter determined to be 79 nucleotides? Is there another gene 79 nucleotides upstream of the start codon?

Response 18. The available DNA sequence for this particular organism is only reported up to the 79th nucleotide upstream the ATG. We incuded this information in the manuscript (lines 358-363).

Comment 19. Line 341 remove bracket.

Response 19. Presumably already removed since we did not detect any extra bracket.

Comment 20. Line 342 What database was used for this recovery of only 8 sequences? A search comparing SBiP1 protein sequence with the NCBI TSA database (taxid 2864) using TBLASTN recovers at least a hundred sequences with >86% identity.

Response 20. For this analysis we did a BLASTP to identify the subcellular localization of the resulting proteins. We excluded some of those sequences under the following rationale: sequences that are duplications, sequences from uncultured organisms, and redundant sequences from sp. data. We are aware that many transcriptome information is available but for our analysis we considered as relevant only the sequences we report here. We included the information in the text to further clarify this (lines 258-264 and 376-378).

Comment 21. Line 347 Some caution should be exercised when discussing gene expression. There is not necessarily any relation between changes in RNA levels and changes in protein levels.

Response 21. We agree and that is why we stated that the results from the transcriptome analysis “suggested” instead of “demonstrated”, that protein synthesis could be synergic” (line 392).

Comment 22. Line 368 A light response may also be just a response to light rather than a circadian behavior.

Response 22. We agree and that is why we make clear at the end of the paragraph that it is only a possibility and a suggestion (lines 413-416).

Comment 23. Line 373 Check spelling of dinoflagellate used.

Response 23. Dinoflagellates is correct but we changed “apicomplexa” to “apicomplexans” to be consistent with the nomenclature used in the sentence (line 383).

Round 2

Reviewer 1 Report

Comments and Suggestions for Authors

The authors have improved the MS in some aspects. but, there are still some issues need to be solved

1. Fig. 1 and Fig. 4 still need provide better resolution.

2. Please chekck the format of the titles of each sections

3. Symbiodiniaceae is the family name, please consider list them in italic.

Comments on the Quality of English Language

The writing of the paper still needs some improvement.

Author Response

All new modifications have been highlighted in magenta.

Comment 1. Fig. 1 and Fig. 4 still need provide better resolution.

Response 2. All figures have 300 dpi resolution and may have lost some resolution when they were incorporated into the manuscript; however, they are fine for publication purposes.

Comment 2. Please chekck the format of the titles of each sections

Response 2. They were modified and formatted according to the journal.

Comment 3. Symbiodiniaceae is the family name, please consider list them in italic. Response 3. Response and reason to not italicize the family name was given in the first response to initial reviewer’s comments but now also please check: a) the reference by the experts in the field of systematics of this family “LaJeunesse et al. 2018, Systematic Revision of Symbiodiniaceae Highlights the Antiquity and Diversity of Coral Endosymbionts. Curr. Biol. 28(16): 2570-2580.e6.”; and b) the reference by “Mathews et al 2024, Multi-Chemical Omics Analysis of the Symbiodiniaceae Durusdinium trenchii under Heat Stress. Microorganisms. 12: 317”, neither of whom italicize the Symbiodiniaceae family name”. Ultimately, we do not have any problem in changing the family name style to italics but we will let the journal set the desired style.   Comment 4. The writing of the paper still needs some improvement.
Response 4. We improved the language although we really did not find significant problems. Please be specific if you require punctual corrections.

.

Reviewer 2 Report

Comments and Suggestions for Authors

I have carefully read though the responses to my previous comments, and am unfortunately still confused about what the authors are actually doing during their analysis of transcriptional regulation. They analyze sequences upstream of the start codon which will include sequences comprising the 5’UTR when transcribed. My specific difficulties with their responses are outlined below.

Response 2

Line 314 talks about analysis using the 50 nucleotides upstream of the start codon, but at least some of this will be found in the transcript as the 5’UTR.

The authors response says “In Symbiodiniaceae however, a large part of the genomic sequence with some regulatory elements close to the ATG initiation codon is conserved in the transcript even after splicing; this may have given the wrong idea that we are analyzing the transcript itself for regulatory elements, but this was not the case.” If the authors are analysing sequence that is conserved in the transcript even after splicing, this is analysing the transcript for regulatory elements.

Response 5

Again, I don’t understand. The authors say “the TTTT motifs found on gene sequences were located between 16 to 24 nucleotides upstream the start codon”. I could not find the length of the 5’UTR but it would seem likely to me it would be longer than 16 to 24 nucleotides. Unless the 5’UTR is less than 16-24 nucleotides the authors seem to be looking at sequence unlikely to be transcriptional regulatory elements.

The authors also say “in dinoflagellates this TTTT motif replaces de canonical TATA-box which is usually located at a similar distance from the start codon in DNA sequences and constitutes the site for transcription initiation.” What is this distance and where does this information come from? Are the authors possibly confusing transcriptional start sites and translational start codons?

Response 7

The authors state “there is no evidence so far, that the spliced leader (SL) sequence is integrated back into the genomic DNA after being added to the mRNA post-transcriptionally “. I do not think this is so. The addition of an SL to the 5’ end of a transcript does indeed happen post transcriptionally, but it has been shown these SL sequences can be recycled back to the DNA in many species (see Slamovitz and Keeling, 2008 Current Biology 18 R550-R552).

When I align the sequences upstream from the ATG (provided in the supplementary data) to the SL sequence, 4 out of 7 sequences seem to show some degree of sequence identity to the SL (Table 1). It is hard to rule out a process akin to what was described by Slamowitz et al. based on this.

Table 1 Similarity of sequences upstream from the ATG to SL

SL             CCGTAGCCATTTTGGCTCAAG             

OP429595       GCAAGGCATTTTTGGTAGAAGCTGAACCTCGAGTGACCATG

CAE7227403      gtagataccttttgattggccggattcATG

OLP91134       tcagagcatttttggtagaagctgaacctcgagtgaccATG

CAE7221339     ccacagctgttttggcaagaagctcaacaggcaagtcagggattct ctgcggtggagtcaaagccATG

CAE7360486                 gatccagtcttttcctttcgtccttagacATG

CAE7932356                 AGAAGGGCTATTTGCTCAACGGCCGTCTGGTCAGGGCTGCAAAA

GTCATTGTCGCCAAAGCGCCGATAATG

Kawagutii        CTCATCCCGTTTTCCAAGTGAAACTCAATTCAACATG

Response 10

Just because the sequence is in the DNA does not automatically confer upon it status as a regulatory element. After all, the 5’UTR is also encoded in the DNA. The authors state that upstream DNA sequence are typically associated with transcriptional regulation, but this would only be true if by upstream the authors mean upstream of the transcription start site. Upstream of the translational start codon includes untranslated sequences unlikely to be involved in transcriptional regulation.

Response 14

Some fraction of the sequence 50 nucleotides upstream of the start codon will be in the 5’UTR. If the authors are unable to determine how much, how can they be sure they are not looking at the 5’UTR? How can they convince the reader the TTTT motif they report  is not part of the 5’UTR?

Response 15

Replace by “C to G”

Response 17

If the authors use 1 kb of sequence upstream of the start codon (line 358), why talk about 50 nucleotides upstream of the start codon (line 312)?

Please check Ref 48, which is used to support replacement of the TATA box by TTTT, as it is for a review on translation.

Author Response

Comments 2, 5, 7, 10, 14 and 17. I have carefully read though the responses to my previous comments, and am unfortunately still confused about what the authors are actually doing during their analysis of transcriptional regulation. They analyze sequences upstream of the start codon which will include sequences comprising the 5’UTR when transcribed. My specific difficulties with their responses are outlined below.

Response 2, 5, 7, 10, 14 and 17. Thank you for your thorough review and insightful comments. After carefully analyzing the points you raised, we realized that there was indeed confusion on our part between the transcriptional start site (TSS) and the translational start codon (ATG). We now realize that your critique was right, and we greatly appreciate it. Given that a comprehensive transcriptional regulation analysis, including promoter analysis, is beyond the scope of this review and requires a more in-depth study, we decided to remove that information from Table 1 and the section of the corresponding analysis (between lines 305 and 306 from the previous version) from the manuscript without significantly modifying the content. We hope this clarifies our intention and aligns the focus of the review more accurately with the subject matter at hand.

Comment 15. Replace by “C to G”

Response 15. No longer relevant as the section where it belonged was removed from the manuscript.

Comment 18. Please check Ref 48, which is used to support replacement of the TATA box by TTTT, as it is for a review on translation.

Response 18. No longer relevant as it was removed from the manuscript, although it does mention the following: “Genomic sequence has now provided evidence that the upstream regions of dinoflagellate genes contain a conserved TTTT motif [17], of particular interest since the dinoflagellates have replaced the TATA-box binding protein (TBP) found in other eukaryotes with a TATA-box like protein (TLF) that binds TTTT instead [21].”

Round 3

Reviewer 2 Report

Comments and Suggestions for Authors

This version adequately addresses my previous concerns regarding analysis of transcriptional regulatory sequences. I fully agree with the authors decision to remove this which is very difficult to analyze in dinoflagellates.

Only one minor comment, remove the reference to TATA in Table 1 legend (line 164).

Author Response

Only one minor comment, remove the reference to TATA in Table 1 legend (line 164).

Response: The TATA box mention on Table 1 legend stating “The details of their TATA-boxes as well as the cis-regulatory elements…” has been changed to “The details of their cis-regulatory elements…”. The new text has been highlighted in green.